# A Decision Support System Using Fuzzy Logic for Collision Avoidance in Multi-Vessel Situations at Sea

**Tanja Brcko *** and **Blaž Luin**

Faculty of Maritime Studies and Transport, University of Ljubljana, 6320 Portoroz, Slovenia;
blaz.luin@fpp.uni-lj.si
* Correspondence: tanja.brcko@fpp.uni-lj.si

**Abstract:** The increasing traffic and complexity of navigation at sea require advanced decision support systems to ensure greater safety. In this study, we propose a novel decision support system that employs fuzzy logic to improve situational awareness and to assist navigators in collision avoidance during multi-vessel encounters. The system is based on the integration of the rules of the Convention on International Regulations for Preventing Collisions at Sea (COLREGs) and artificial intelligence techniques. The proposed decision model consists of two main modules to calculate the initial encounter conditions for the target vessels, evaluate the collision risk and navigation situation based on COLREG rules, sort the target vessels, and determine the most dangerous vessel. Fuzzy logic is used to calculate the collision avoidance maneuver for the selected ship, considering the closest point of approach, relative bearing, and the ship's own speed. Simulation tests demonstrate the effectiveness of the fuzzy-based decision model in scenarios with two ships. However, in complex situations with multiple ships, the performance of the model is affected by possible conflicts between evasive maneuvers. This highlights the need for a cooperative collision avoidance algorithm for all vessels in high traffic areas.

**Keywords:** multi-ship collision avoidance; fuzzy reasoning; decision support model

## 1. Introduction

The latest approaches to ensuring greater safety at sea are reflected in the form of decision support systems that harness advanced computer technology to enhance situational awareness and decision-making both onboard the ship and ashore. The main functions of the systems include tools such as prediction of the ship's course, warnings of possible ship collisions, groundings and approach to a guard zone, planning of collision avoidance maneuvers based on COLREG rules (Convention on the International Regulations for Preventing Collisions at Sea, 1972), etc. Such decision support can be found to some extent in ECDIS (Electronic Chart Display and Information System) on a modern ship today.

As the traffic density increases, navigators face a distinct challenge when it comes to avoiding collisions in multi-vessel situations. Existing COLREG rules governing the right-of-way between vessels in close quarter situations do not address such scenarios, leaving the decision-making responsibility in the hands of the navigators. This becomes particularly problematic at sea, where ships, especially in busy areas, lack sufficient time to communicate and coordinate collision avoidance measures. Conversely, the rules regulate collision avoidance for various types of encounters involving two vessels, which can potentially be applied to situations involving multiple vessels. However, to do so effectively, it is necessary to classify and determine the navigational situation. By employing decision-making tools, this classification process becomes simple and efficient, particularly if the tool is integrated into a navigation system utilized daily.

The COLREG rules consider several factors in determining the right of way: the area of navigation where the vessels are located, the relative position of the vessels involved,

the type of the navigational situation (head-on, crossing, overtaking), and the navigational status of the vessels (power-driven, fishing boat, sailing boat, restricted in her ability to maneuver, etc.). From a navigator's point of view, the first important step in a situation where several vessels must avoid a collision is to identify the most dangerous vessel among them, followed by the choice of a maneuver that meets the safety requirements of the COLREG regulations.

### 1.1. Literature Review

Many researchers deal with decision models, some solve them holistically, with heuristic algorithms such as path planning, while others solve individual steps of collision avoidance, i.e., the extent of a speed or course change, which are considered deterministic algorithms [1]. However, not all studies consider COLREG as a part of the decision. The following literature review mainly focuses on three basic components of decision models: the Collision Risk Analysis, the Navigation Situation Classification, and the Collision Avoidance Maneuver.

### 1.1.1. Collision Risk Calculation Algorithms

The collision risk analysis component includes two functions of the model: namely, detecting the risk of collision with the target vessel and determining the right-of-way between vessels. The quantitative methods for calculating the collision risk could include, first and foremost, the calculation of the DCPA (Distance to the Closest Point of Approach). The Closest Point of Approach (CPA) is the point at which two ships will meet closest to each other. The smaller the distance to this point (DCPA), the greater the risk of collision. This is the first indicator that shows the possibility of collision or crossing the safety domain of one's own vessel.

A vessel's safety domain is an area around a vessel that must remain free of other vessels and fixed installations. In practice, this area takes the form of a circle, the radius of which is subjectively determined or established in advance by the ship's safety management system. The collision avoidance rules basically do not specify how large a vessel's safety domain should be; but, according to Cockcroft [2], it should be limited to two NM (nautical miles) in poor visibility or may be even smaller at low speeds in heavy traffic or when overtaking.

Through various developments in the calculation of the collision risk, the safety domain has also evolved, both in size and shape. Most of the early developments were based on statistical and analytical methods, and the domains were usually oval or elliptical in shape. One of the first was Goodwin [3], who took the COLREG rules as a basis and proposed a division of the navigation area into three sectors corresponding to the angles of the ship's navigation lights. The model was based on a statistical analysis of data from numerous registers and simulations.

Dynamic models of polygonal shapes are found in conjunction with the use of artificial intelligence techniques: Pietrzykowski [4] presented a fuzzy ship domain (combined with a self-learning neural network) as a safety criterion in offshore navigation; a similar method was used by Wang [5], who wrote that fuzzy boundaries in the ship domain are more practical for navigators than well-defined boundaries in assessing navigational safety. Su et al. [6] used variables in their fuzzy system to calculate the size of the safe encounter domain: relative approach speed, size of both ships, and sea state. Later, the development of the domains was dynamically adapted to the different navigation situations: size of vessels, traffic density, relative speeds, type of navigation situation, weather conditions, visibility, and more [7–9]. Du [10], on the other hand, presented an empirically determined ship domain based on a large dataset of ship encounters detected from AIS (Automatic Identification System) data. AIS data today are a very good source for the study of maritime traffic, since it works with dynamic ship data such as position, dimensions, speed, course, etc.

The authors who have dealt holistically with the problem of avoiding collisions at sea have usually used, in the model for the safe area of the ship, a simple radar circle with a radius of up to one NM [11–13]. Li [14] adapted two NM as minimum DCPA and Hu [15] set the value of the safety ship domain 12 to 14 times the ship length. Some dealt with simple ellipses where the size was determined by the length and the width of the ship [16,17]. The quadratic ship domain was used in [18] as the safety area of the ship, with the size determined by four radii (i.e., forward, aft, right, and left). Many authors did not specifically define the size of the domain in the decision models, including [19–21], or they chose a minimum passing distance [22,23]. In the maritime industry ship's domain, research has primarily enabled the application of new domains in existing navigation devices such as radar systems and electronic charts to enable more accurate collision risk calculations.

### 1.1.2. Algorithms for Determining Collision Risk and Right of Way in Complex Vessel Encounters

Determining the risk of collision and establishing the right of way when two ships encounter, especially in the open sea, is not a significant navigational hurdle. Of greatest importance are the values of the DCPA, the TCPA (Time to Closest Point of Approach), and the relative position of the vessels, as these form the basis for collision avoidance maneuvers.

In areas with dense traffic, there is a higher probability of encountering complex situations involving multiple vessels simultaneously [24]. The more complex the situation, the less situational awareness navigators managing the vessels may have, as their own vessel can be in both a give-way and stand-on position at the same time. This is influenced not only by the vessel's status but also by the navigational area in which the vessels are located (narrow straits, traffic separation scheme, open sea, etc.).

According to COLREG rules 16 and 17, the "Stand-on" vessel must maintain its course and speed, while the "Give-way" vessel must alter its movement to avoid impeding the vessel with the right of way. As mentioned earlier, COLREG rules do not cover complex situations; hence, the need for finding solutions becomes even more significant and demanding. Part of the collision avoidance process in complex scenarios is also the identification of the situation: which vessels are at risk of collision, determining the right of way with each vessel, and to identify the most dangerous vessel among them.

Some authors have dealt with this identification. Zhuo [25] presented an algorithm for calculating the time at which an avoidance maneuver should be initiated and the time frame in which a vessel should take action to identify the most dangerous vessel in a multi-vessel encounter. Elements that influence the time value are DCPA, TCPA, the dynamics of the target vessel, the maneuvering characteristics of the own vessel, the course of the target vessel, and the sea state. To calculate the above values, an adaptive self-learning system was used in combination with neural networks and fuzzy logic techniques. Hasegawa et al. [26], who addressed the problem of a multi-ship collision avoidance, presented the calculation of the collision risk (CR) using fuzzy logic. He used DCPA and TCPA values as input data of the fuzzy inference system and the CR value as output decision. A target ship with the highest CR value was a stand-on ship. Hu [15] followed a similar approach and used, for the input parameters, relative distance, bearing, and speed in addition to DCPA and TCPA. Zhang [27] used the speed ratio between two ships instead of the speed of the ship. Bukhari et al. [28] used DCPA and TCPA values as input data and added the value of VCD (Variation of a Compass Direction), which indicates the change in bearing of a target ship over time. The output decision was the degree of collision risk, and the vessel with the highest value was a stand-on vessel. Wang [29] adopted a basic CR calculation method to construct risk membership functions of DCPA and TCPA, which was also used by Zheng [30]. Ahn et al. [31] focused on situations with limited visibility where he calculated the CR using neural networks. As input variables, he used the speed, the course of the own

and the target ship, the distance between the ships, the bearing of the target ship, and the safety range of the ship.

### 1.1.3. Algorithms for Calculating Collision Avoidance Maneuvers for Multiple Ship Encounters

The next step in the process of collision avoidance in complex scenarios is the planning of collision avoidance maneuvers. The different approaches by researchers to the problem are mainly reflected in the choice of trajectory planning strategies influencing the multi-ship collision avoidance maneuver. Here, we find various algorithms using methods such as deep reinforcement learning, fuzzy logic, artificial potential field, neural networks, swarm intelligence, collision-free trajectory generation, model predictive control, and so on. The main challenge for researchers is to incorporate COLREG rules into decision models, taking into account that they are adapted as a fundamental part of their designs [32]; therefore, studies that do not contain COLREG rules are excluded in the literature review. Liu [33] presented an algorithm for determining the direction of the collision avoidance maneuver, in which a course change amplitude of 10° was chosen and the maximum turning course was set at 60°. Various parameters were considered, such as the distance between the two vessels, true bearings, relative bearings, relative speed, and the heading of the target and the vessel. Lu's [34] work combined the artificial potential field method with a collision avoidance algorithm executed by a particle swarm optimization algorithm, while Miao's [35] work used an improved hybrid A* algorithm that searches for appropriate motion options. Both articles presented a method for calculating the corresponding change in motion of the ships involved in a collision avoidance situation. Zhang [27] introduced an enhanced approach that combines the Velocity Obstacle method, model predictive control, and a ship trajectory prediction model. The objective of this method is to determine a viable space for collision avoidance maneuvers while considering the COLREG, and the constraints imposed by the ship's maneuverability. Authors [20,30], on the other hand, used a proximal policy optimization algorithm for collision avoidance path planning in multi-ship scenarios.

Several authors have also proposed a collision avoidance maneuver based on fuzzy logic. A fuzzy logic algorithm was used as the basis for calculating collision avoidance (course change and speed reduction) using three input parameters: the relative and encounter angle in a ship encounter, and the value of collision risk [36]. Perera [19] gave five input parameters: the region where the target ship is located, the relative course of the target, the degree of encounter risk, the distance to the target ship, and the relative speed of approach. Based on these parameters, the model decided the need to change course or speed based on COLREG rules. The selection of the navigation strategy in the traffic separation scheme, using a decision model based on a fuzzy logic algorithm, was also proposed by Wu [37], who analyzed the dynamic characteristics of the navigation process. Using a similar fuzzy logic approach, the risk of collisions with static and moving objects was calculated by Wu [38] and Hu [15].

In a previous article [39], the authors presented a multi-parameter decision model for collision avoidance at sea using fuzzy logic in the situation of two ships' encounter. In the article, the structure and operation of the fuzzy inference system, decision validation, and examples of model tests were presented. Building upon this prior research, the present article shifts its focus towards the intricacies of ship maneuvering characteristics and collision avoidance strategies in scenarios featuring multiple vessels at sea.

The paper is structured into several sections, each addressing a specific aspect of the research. The first section is the "Introduction," which provides an overview of the latest approaches in ensuring safety at sea using artificial intelligence-based decision support systems. The introduction also addresses existing research on decision models for collision avoidance at sea, algorithms, and methods for calculating collision risks and maneuvers in multiple vessel scenarios. Section 2 is the "Methodology", which outlines the systematic approach used in the paper. Section 3 presents the decision model, which is the core of

the research. It explains the simulation of expert decision-making for collision avoidance at sea. The following section provides a practical demonstration of the decision model in a multi-ship scenario. It shows how the model calculates which is the most dangerous ship, determines the appropriate time interval for collision avoidance, and suggests an avoidance maneuver. Section 4 is the "Simulations", where the model's performance is evaluated through simulations in various multi-ship scenarios. The authors utilize a simplified nautical simulator coupled with the fuzzy collision avoidance system to assess the efficiency of the proposed decision model. The fifth section is the "Discussion", where the authors analyze the results of the simulations and discuss the strengths and limitations of the proposed decision model. Conclusions and suggestions for future research are presented in the sixth section.

## 2. Methodology

The methodology of the paper consists of several approaches:

1.  Literature Review: The authors conducted a literature review of scientific papers to gather information on emerging decision models for collision avoidance at sea. Special attention was on articles that addressed multi-vessel situations and incorporated the use of COLREG rules in their models. The review mainly focused on two components of decision models: the collision risk analysis component (CR) and the collision avoidance maneuver component (CA).
2.  Algorithm Development: The authors developed algorithms for calculating collision risk and collision avoidance maneuvers in multi-ship encounters. They considered various parameters, such as the DCPA, TCPA, relative bearings, relative speed, ship types, and navigation area, to assess the collision risk and determine the right-of-way between vessels. A classification algorithm was partly presented in the conference paper [40].
3.  Fuzzy Logic: Fuzzy logic was utilized as a decision-making tool in the collision avoidance system. The authors implemented fuzzy inference systems with triangular or trapezoidal membership functions to determine the degree to which inputs belonged to different fuzzy sets. The fuzzy logic approach was used to calculate collision avoidance maneuvers in multi-ship encounters.
4.  Simulation: A Monte-Carlo class of simulations, involving numerous runs to evaluate the performance of the proposed decision model for collision avoidance in multi-ship scenarios, was conducted. The simulations were carried out using a simplified ship dynamics simulator coupled with the fuzzy collision avoidance system. Different initial positions, speeds, and orientations were considered for each ship to assess collision avoidance performance in various scenarios.

## 3. Decision Model

The aim of the decision model is to simulate the decision-making of experts in avoiding collisions at sea. The knowledge that the model must contain is summarized in a multi-parameter decision model scheme consisting of two modules (Figure 1):

MODULE 1 "Initial parameter" calculates the initial conditions for the encounter of ships based on the data of ship targets and the ship.

MODULE 2 "Decision Model" is divided into two main components:

*   Component 1 "Collision risk assessment and navigation situation analysis",
*   Component 2 "Collision course maneuver calculation".

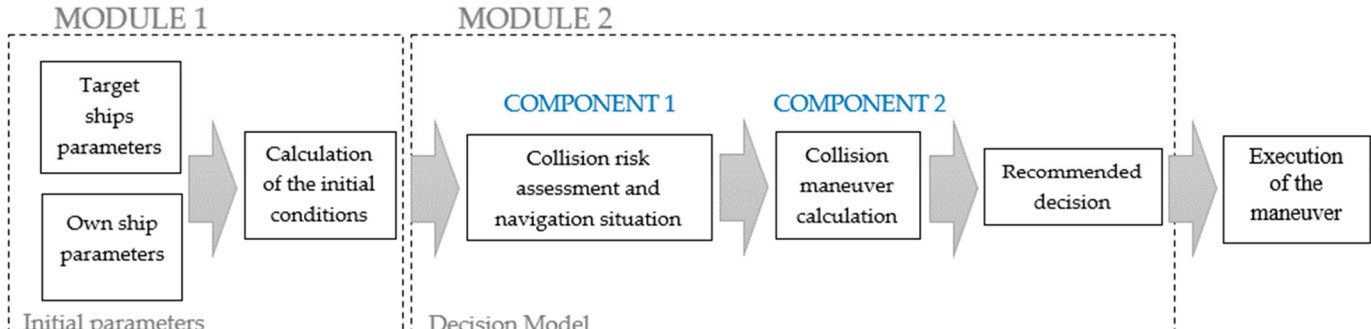

**Figure 1.** Decision model scheme.

Component 1 is based on the operation of the sorting algorithm and the use of COL-REG rules, and component 2 includes fuzzy reasoning in addition to these rules, which simulates the knowledge of the expert. The operation of both components is explained in more detail below.

### 3.1. Component 1—Collision Risk Assessment and Navigation Situation

Component 1 contains the algorithm for determining the most dangerous ship (Figure 2). The inputs used are DCPA, TCPA, direction and speed of the target ship, ship type, and COLREG rules eight, nine, 10, 13, 14, 15, 16, 17, 18, and 19. In the "parameter processing" step, the algorithm collects and analyzes the navigation input parameters of the target vessels (initial conditions) within the desired range. The parameters observed are the bearing, range, DCPA, TCPA, and ship's type. In the next step, the algorithm sorts the vessels according to the DCPA value—for further processing, select the vessels whose DCPA value is smaller than the vessel's safety domain (depending on the navigation area and meteorological/oceanographic conditions). If there is only one target in an area, the algorithm determines the right-of-way according to COLREG rules and suggests an evasive maneuver if necessary. If there are multiple vessels, the right-of-way for each of the vessels is determined. The target ships that have right-of-way are sorted by the algorithm according to the minimum TCPA. The ship with the lowest TCPA is selected for avoidance. Two exceptions are used in the algorithm:

- Exception 1—Any target ship within two NM or less, regardless of position or status, has the right of way (as per COLREG rule 17).
- Exception 2—If two vessels in sectors I or IV (see Figure 3) have the right of way, the vessel that is closer should be avoided.

In the algorithm, 90 conditional sentences are used to determine the right-of-way according to the COLREG rules. Their structure is presented in Table 1.

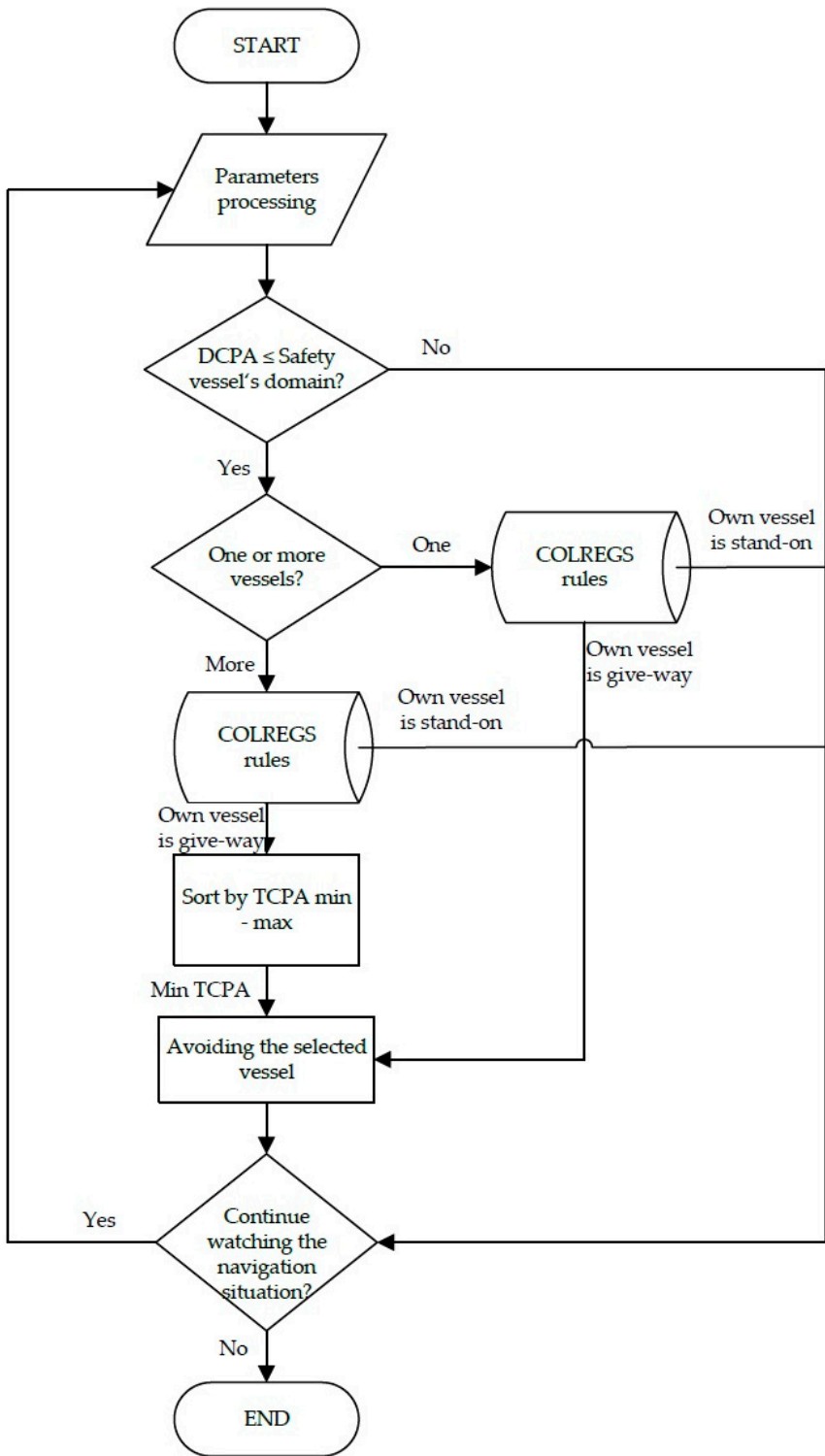

**Figure 2.** Algorithm for determining the most dangerous ship, adapted from: Brcko [40].

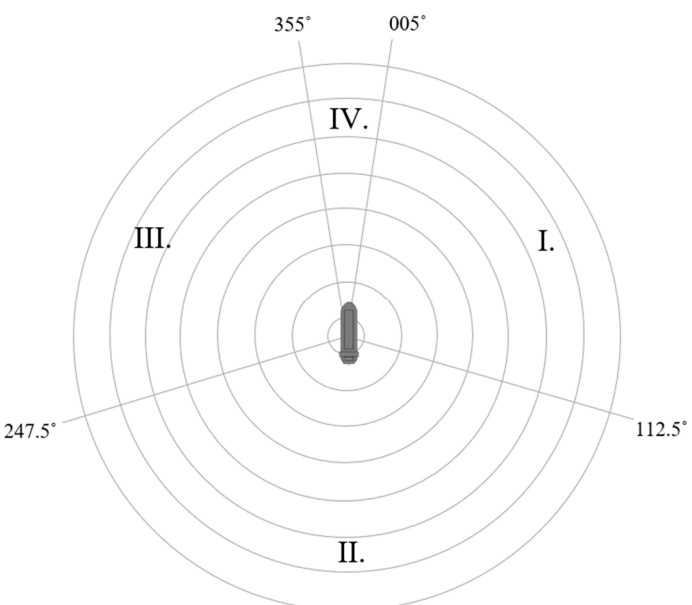

**Figure 3.** Sectors of the vessel's own domain.

**Table 1.** Structure of the algorithm.

| | | | |
|---|---|---|---|
| if | A | Navigation area | Open sea<br>Traffic Separation Scheme<br>Narrow Channel |
| and | B | Position of the target vessel by sector | Sector I<br>Sector II<br>Sector III<br>Sector IV |
| and | C | Type of navigation situation | Crossing<br>Head-on<br>Overtaking |
| and | D | Navigational status of the target vessel | Power-driven vessel<br>Moored<br>Not under command<br>Restricted in her ability to maneuver<br>At anchor<br>Constrained by her draught<br>Aground<br>Engaged in fishing<br>Sailing vessel |
| then | E | The right of way | The target vessel has the right of way or<br>The own vessel has the right of way |

The navigation area is divided into the open sea, the traffic separation scheme, and the narrow navigation channel. From the ship's own perspective, the target ship may be in any of the four sectors (Figure 3), depending on their relative bearings (Equations (1)–(4)):

$$P_t \sec I = 5° \leq RB \leq 112.5° \tag{1}$$

$$P_t \sec II = 112.6° \leq RB \leq 247.5° \tag{2}$$

$$P_t \sec III = 247.6° \leq RB \leq 354.9° \tag{3}$$

$$P_t \sec IV = 355° \; \leq \; RB \; \leq \; 004.9° \tag{4}$$

where $P_t$ stands for the target position and RB is the relative bearing of the target. The boundaries of sectors I–III are defined with COLREG rules. The type of navigational situation encountered (rules 13, 14, or 15) is determined by conditional statements. These rules apply only when there is a risk of collision.

According to COLREG rule 18, ships have different priorities over other ships depending on their navigational status. This rule replaces rules 14 and 15, which apply only in situations where there is a risk of collision between power-driven vessels.

The responsibility of one's ship changes with the different navigation conditions. The algorithm provides one of two possible choices: the own ship is a give-way or a stand-by ship. According to the COLREG rules, the give-away ship should maintain its course and speed, while the stand-by ship must perform a collision avoidance maneuver.

### 3.2. Component 2—Collision Course Manoeuvre Calculation

Collision avoidance is a process that requires planning and observation of a dynamic navigational situation over a reasonable period of time. Path planning is extrapolating the trajectory of ships with a time delay. The paper suggests that the collision avoidance maneuver is calculated based on the data of the most dangerous ship. At the same time, the model checks the risk of collision with all the ships involved and plots the trajectories of the ships. Finally, the model calculates a time frame within which collision avoidance is safe and in accordance with COLREG rules.

Determining the avoidance maneuver is a two-step process. First, the model collects input data of the ship's own and target vessels to assess the collision risk observing minimum acceptable TCPA and DCPA parameters under the assumption speed and the course remains constant. The calculation using relative positions and speeds is defined by the Equations (5) and (6), where Xt and Yt represent the relative position coordinates of the target vessel, while Vrx and Vry denote the components of the relative velocity vector. Vr stands for the relative velocity of the approaching vessels [41].

$$DCPA = \left| \frac{(X_t \cdot V_{ry}) \; - \; (Y_t \cdot V_{rx})}{V_r} \right| [M], \tag{5}$$

$$TCPA = - \; \frac{(Y_t \cdot V_{ry}) + (X_t \cdot V_{rx})}{V_r^2} \cdot 60 \; [min], \tag{6}$$

The collision risk is directly dependent on the DCPA and TCPA and appropriate maneuvers are chosen according to the Fuzzy rules without intermediate quantification of the risk as it was done during the evaluation phase as defined by the Equation (11).

To predict the target vessel position at the time of the collision avoidance maneuver, the time delay calculation is used to obtain the new relative position of a target vessel. At this point four parameters are calculated for further processing as input variables in a fuzzy inference system [39]:

- DCPA—Distance to Closest Point of Approach,
- AP—Action Point distance to the target vessel,
- RB—Relative Bearing of a target vessel,
- Vo—Own vessel Velocity.

The fuzzy inference system (FIS), also known as the rule-based fuzzy system, is the process of formulating the mapping from a given input to an output using fuzzy logic (Figure 4).

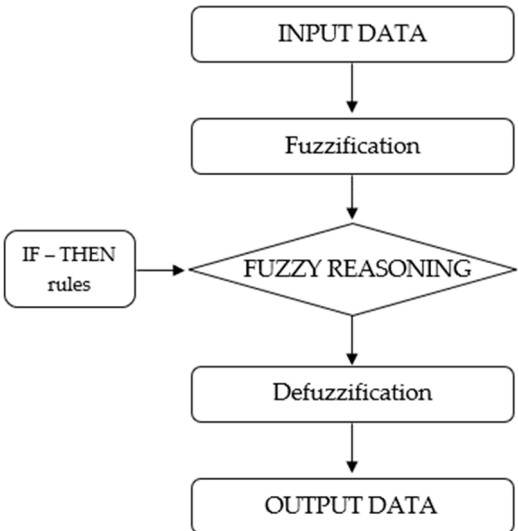

**Figure 4.** The structure of a fuzzy inference system.

It is a main element of the fuzzy logic system. The FIS formulates rules and based on these rules, the decision is made. The FIS type in this paper is "Mamdani", which is the most used fuzzy method. The first step is to take the inputs and outputs and determine the degree to which they belong to each of the corresponding fuzzy sets using triangular or trapezoidal membership functions (Figures 5–9). A fuzzy system is a set of fuzzy rules that convert fuzzy inputs into fuzzy outputs. It consists of a rule-based system of IF (antecedent)—THEN (consequent). A total of 216 rules forms the IF-THEN statements (Table 2).

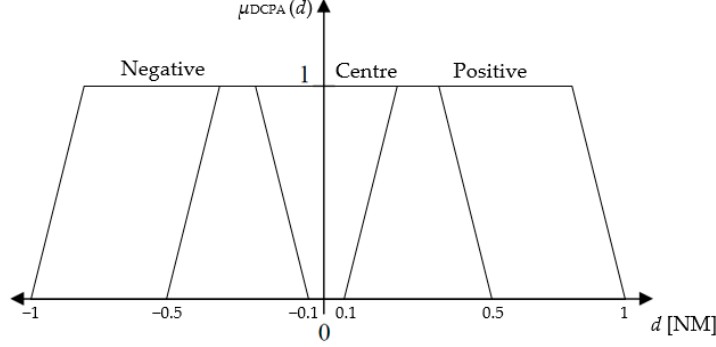

**Figure 5.** Fuzzy membership functions of DCPA parameter.

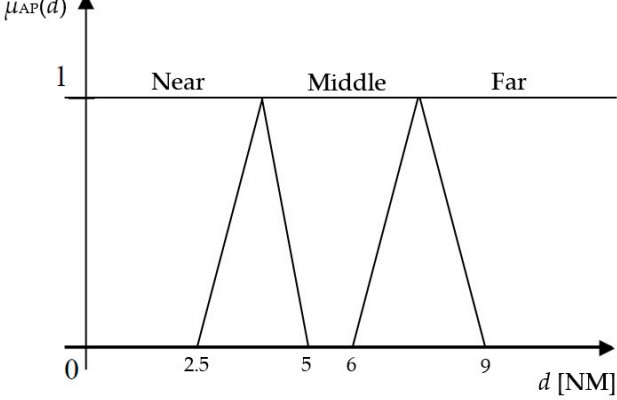

**Figure 6.** Fuzzy membership functions of AP parameter.

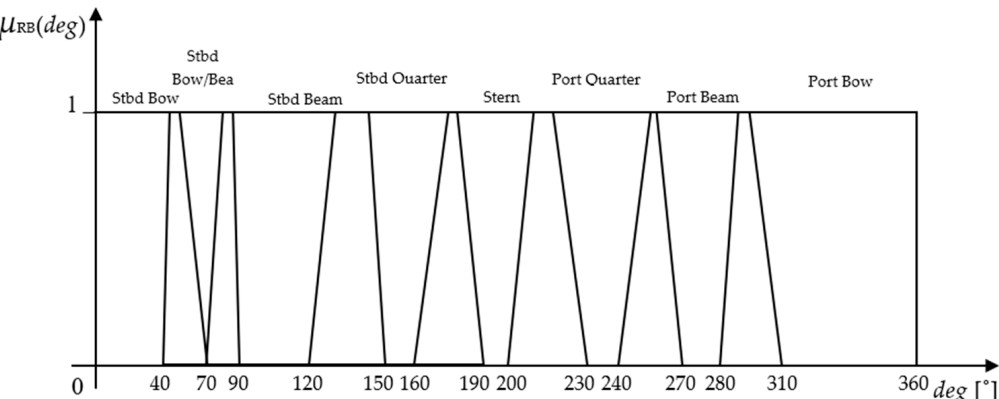

**Figure 7.** Fuzzy membership functions of RB parameter.

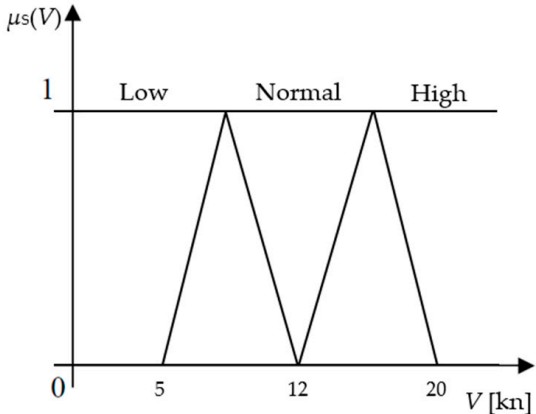

**Figure 8.** Fuzzy membership functions of the vessel's velocity parameter.

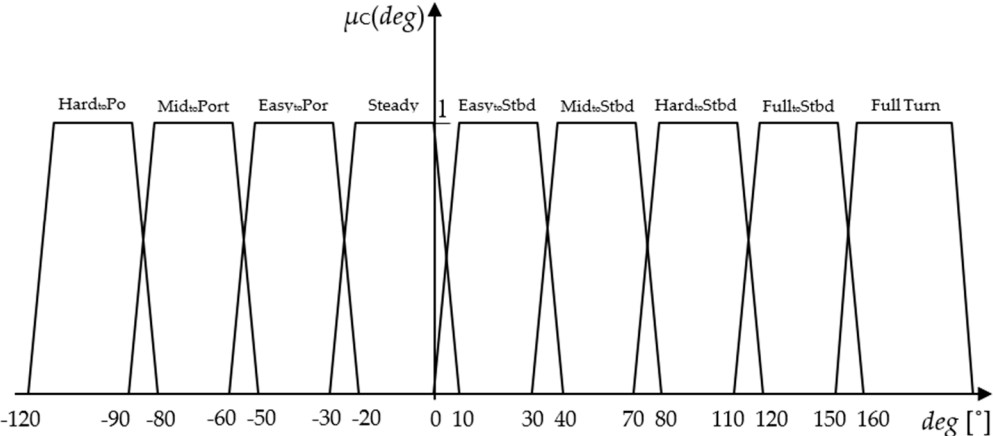

**Figure 9.** Fuzzy membership functions of the output parameter, the alteration of the vessel's own course.

**Table 2.** IF–THEN statements.

| IF | DCPA | Negative, Positive, Center |
|---|---|---|
| AND | AP | Near, Middle, Far |
| AND | RB | Stbd Bow, Stbd Bow/Beam, Stbd Beam, Stbd Quarter, Stern, Port Quarter, Port Beam, Port Bow |
| AND | V | Low, Normal, High |
| THEN | Course alteration | Steady, Easy to port/starboard, Mid to port/starboard, Hard to port/starboard, Full to starboard, Full turn |

Source: Adapted from Brcko et al. [39].

The second step involves the intricate computation of the decision or course alteration, employing the advanced principles of fuzzy logic methodology. In this phase, the distinguished techniques of the bisector and centroid are employed to defuzzify the output function derived from the fuzzy system. The bisector method strategically deploys a vertical line, effectively partitioning the region into two distinct sub-regions, each with an equal area. Although not universally so, it often coincides with the centroid line, which holds paramount importance in the realm of the Mamdani's Fuzzy Inference Systems (FIS) technique. The centroid, characterized as the center of gravity, represents the prevailing and most widely adopted approach within Mamdani's FIS method. Its pivotal role in decision-making is derived from its ability to effectively balance the distribution of fuzzy sets, providing a reliable and well-founded basis for subsequent actions.

Upon the completion of this comprehensive decision model, the ultimate outcome materializes as the precise alteration of the vessel's course, precisely expressed in degrees, either towards the port or starboard side.

### 3.3. Operation of the Model on an Example

In a multi-vessel collision avoidance situation, the model calculates the parameters DCPA, TCPA, position of the CPA point, RB, relative speed, and relative course based on the initial data for each vessel. Then, the right-of-way is determined according to the COLREG rules and the most dangerous vessel (which has the right-of-way) is selected. Based on the parameters of the most dangerous vessel, the collision avoidance course is calculated for each two-minute time delay. During this process, the model observes the DCPA parameters of all ships. Finally, it calculates a time interval in which collision avoidance is recommended, considering all safety parameters and recommending a maneuver. The simulation tests the fuzzy logic response for the encounter situation of three ships in sectors I and III, governed by COLREG rule 15 (crossing). The simulation observes the tuning of the set parameters and rules of the fuzzy inference system that follows COLREG rule 15.

Table 3 shows the initial parameters of the target ships in the radar diagram, where "C" is the ship's course, "V" is the ship's speed, "dt" is the distance to the target ship, and "ωt" is the bearing of the target ship. All ships in the simulation are underway using power.

**Table 3.** Initial parameters, zero min.

|  | Own Ship | Target 1 | Target 2 |
|---|---|---|---|
| C [°] | 225 | 160 | 338 |
| V [kn] | 20 | 17 | 13.5 |
| dt [NM] | - | 8 | 7 |
| ωt [°] | - | 270 | 205 |
| Navig. status | Power driven | Power driven | Power driven |

In the next step (Table 4), the model calculates the initial navigation conditions, the most important data being DCPA, TCPA, the position of the CPA point with respect to its

own ship (Figure 10), and the relative bearing of the target ship. Next is the determination of the right-of-way for each ship (Table 5).

**Table 4.** Collision risk assessment with targets 1 and 2.

|  | **Target 1** | **Target 2** |
|---|---|---|
| DCPA [NM] | 0.731 | 0.754 |
| TCPA [min] | 23.9 | 14.8 |
| CPA position | negative | positive |
| RB [°] | 45 | 340 |

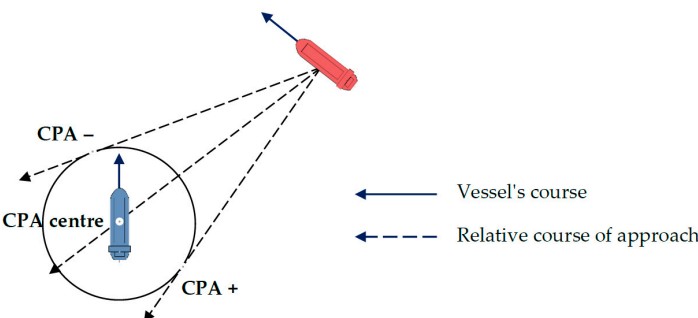

**Figure 10.** Position of Closest Point of Approach (CPA) for a different relative course of approach, adapted from: Brcko et al. [39].

**Table 5.** Determination of the right of the way with targets 1 and 2.

|  | **Target 1** | **Target 2** |
|---|---|---|
| Sector | I | III |
| Vessel's type | Power driven | Power driven |
| Nav. Situation | Rule 15 | Rule 15 |
| Vessel with the right of the way | Target vessel | Own vessel |

For each minute of the time delay, the model calculates the relative position of the target ships. Table 6 shows the position of the ships in the sixth minute of observation. Based on these data, the model calculates the course change shown in Table 7 and reassesses the navigation situation by calculating the DCPA and TCPA for each ship (Table 8).

**Table 6.** A new relative position of a target vessels for a time delay of six min.

|  | **Target 1** | **Target 2** |
|---|---|---|
| $\omega t$ [°] | 268.25 | 209.13 |
| dt [NM] | 6.01 | 4.21 |
| RB [°] | 43.25 | 344.13 |

**Table 7.** Calculated input parameters for the fuzzy inference system (FIS).

| DCPA [NM] | $-0.73$ |
|---|---|
| AP [NM] | 6.01 |
| RB [°] | 43.25 |
| V [kn] | 20.00 |
| Course alteration [°] | 47.5 |

**Table 8.** Reassessment of the risks of collision.

|  | Target 1 | Target 2 |
|---|---|---|
| DCPA [NM] | 3.4367 | 1.6389 |
| TCPA [min] | 9.6 | 12.3 |
| Vr [kn] | 30.8 | 18.9 |
| Cr [°] | 123.1 | 52.0 |

The sample simulation shows a close quarter situation with two ships, where the target ship one has the right of way and the target ship two has to perform a collision avoidance maneuver (Figure 11).

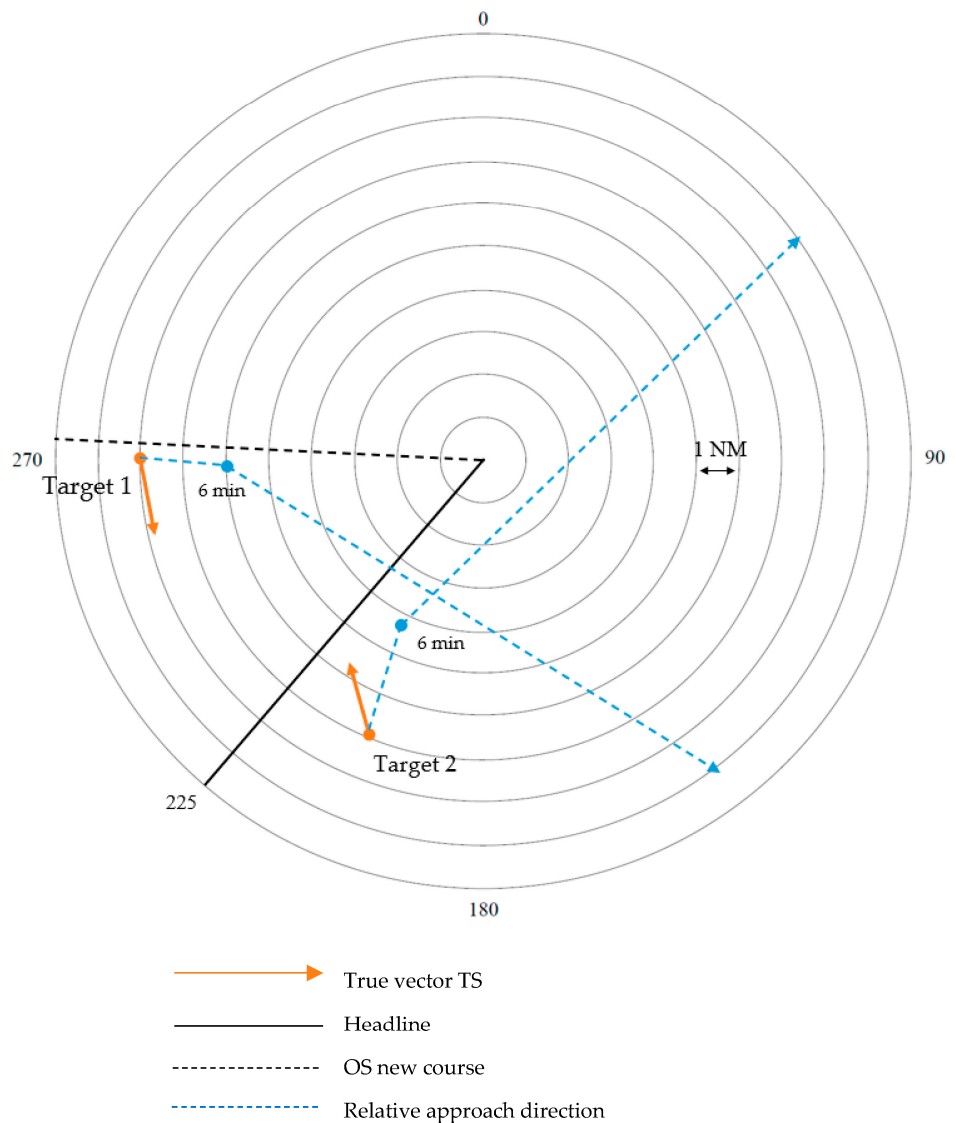

**Figure 11.** Collision avoidance maneuver in the radar diagram.

The maneuver is calculated based on the parameters of ship one. If ship two violates the COLREG rules, the appropriate time to start the maneuver is between two and nine minutes after the start of the observation. Otherwise, the choice of maneuver is considered appropriate until the 19th minute of the time delay.

## 4. Simulations

Since the fuzzy system has been well evaluated using two ship scenarios in previous work [39], this article extends its use to cases with multi-ship scenarios. The evaluation of complex navigational situations, when a collision avoidance maneuver may cause a hazardous situation with another vessel in its proximity, can be evaluated by repeating the simulation many times using different initial positions, speeds, and directions. Doing this with conventional nautical simulators would be extremely time-consuming as thousands of simulation runs are required. This approach is known as the Monte-Carlo simulation. To carry it out, a simplified nautical simulator has been developed based on the ship simulator UTSeaSim, Version 1.0, October 2013 [42], modified to include the impact of the rudder direction on longitudinal speed, speed, and direction control, and was tuned for the simulation of larger vessels. It assumes that ship longitudinal motion is described by

$$\ddot{x} m = \sum_i F_i \tag{7}$$

where $x$ is ship position, $m$ is the displacement, and $F_i$ is the forces acting on the ship, such as propulsion and resistance forces. Ship rotational motion is defined by the

$$\alpha I = \sum_i \tau_i \tag{8}$$

where $\alpha$ is angular acceleration, $I$ is the moment of ship inertia, and $\tau_i$ are moments acting on the ship due to rudder force and lateral resistance forces. Further details of the model can be found in the publication and code of the model authors [42].

To control the speed and course, PID regulators were used to model the autopilot; therefore, the engine speed command is a function of

$$throttle = c_1 \left( v_{target} - v_{ship} \right) + c_2 \left( a_{target} - a_{ship} \right) + c_3 v_{target} \tag{9}$$

where *throttle* is the engine speed command based on regulator constants c and differences between the target and actual speed, target acceleration $a_{target}$, and actual acceleration $a_{ship}$.

In a similar manner, rudder command is controlled by

$$rudder = d_1 \left( \theta_{target} - \theta_{ship} \right) + d_2 \left( \dot{\theta}_{target} - \dot{\theta}_{ship} \right) \tag{10}$$

where $\theta$ is ship heading and $d_i$ are regulator constants.

The model has been coupled with a fuzzy collision avoidance system that determines avoidance maneuvers for two-ship interactions. When there are more than two ships, the target ship for the avoidance maneuver has been chosen by the lowest TCPA and by observing the priority according to the COLREG rules in the open sea (see Section 3.1). Basically, the evasion of approaching ships coming from the right, overtaking, and head-on avoidance were implemented. In the simulation, all the ships had the same status and none of them had restricted maneuverability.

The approach is illustrated in Figure 12, where the simulator provides the fuzzy collision avoidance system with positions, speeds, and relative bearings of other vessels. Once a need for collision avoidance maneuver arises, the system outputs two commands: heading offset and speed decrease if needed. Afterwards the autopilot system adjusts the heading and speed according to the commands by outputting the desired rudder angle and throttle position which are forwarded to the ship model that is impacted by the wind, sea current, and wave conditions.

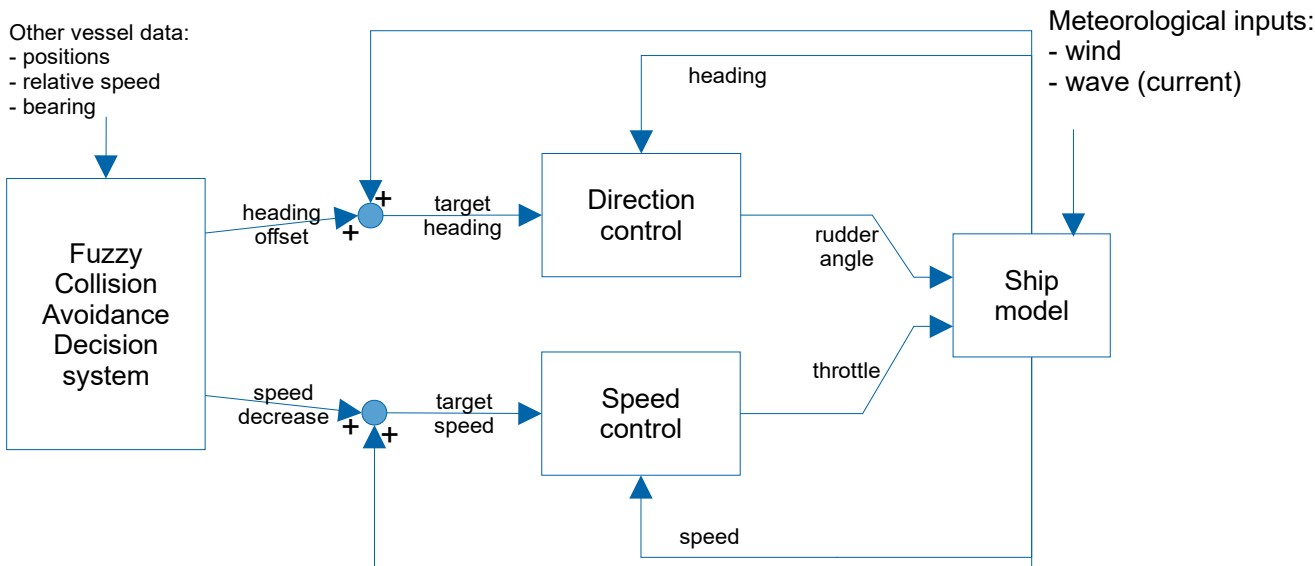

**Figure 12.** Simulation of the collision avoidance system.

Every ship in the system was controlled by a simulated autopilot coupled to the same fuzzy system as used before. Ship particulars for all ships in all scenarios were:

- Length: 100 m
- Breadth: 18 m
- Displacement: 2500 t
- Power output: 4000 kW

The performance indicators that evaluate the maneuvers are the minimum distance, maximum encounter risk, and maximum average encounter risk of a simulation case. To assess the encounter risk, a modified estimation approach was utilized, building upon the work of [43,44]. This modification ensures that the resulting risk is not influenced by vessel proximity when the distance is already increasing (i.e., when TCPA is negative). Therefore, the encounter risk R can be calculated according to Equation (7):

$$R = \begin{cases} e^{-|DCPA|} \cdot e^{-6TCPA}, TCPA \geq 0 \\ \qquad\qquad 0 \qquad\quad, TCPA < 0 \end{cases}. \tag{11}$$

In order to focus only on evasion maneuvers, meteorological conditions were set as neutral in all simulation scenarios. The air and water temperature was 20 °C with a zero water current, wave height, and wind velocities. Simulations were carried out by setting the ships' initial positions on a circular ring area with a specified internal and external radius and heading directed towards the center of the ring. An example of three ship scenarios is shown in Figure 13, where the red area marks the possible ship initial positions. They are obtained by generating a random radius between the lower boundary r1 and upper boundary r2 and a random orientation angle α for each ship.

Each of the scenarios was repeated 1000 times to obtain a collision avoidance performance at different initial positions, speeds, and orientations.

The results in Table 9 and Figure 14 indicate that the model performed flawlessly during the reference simulation with only two vessels (case eight), which was carried out to determine the reference risk for the described methodology. During two-ship simulations, no cases of possible collisions were recorded. As the number of ships involved in a simulation increases, more critical conditions arise. One parameter that severely impacted the performance of the maneuvers is the minimum initial distance between two vessels. If there was not such a limitation, initial positions could be set in a way that avoidance could be physically impossible no matter how the system responds.

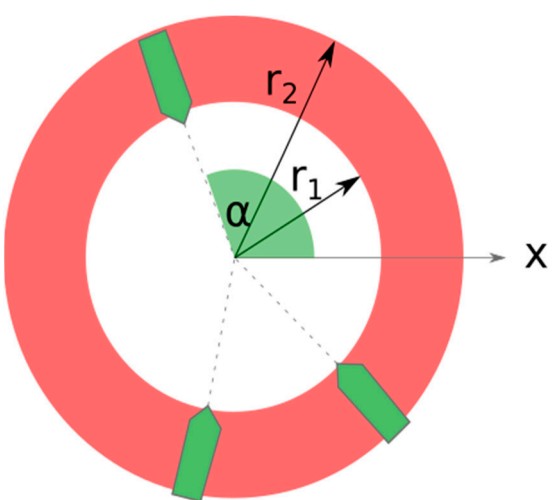

**Figure 13.** Initial positions at the beginning of a simulation run.

**Table 9.** Description of the simulation runs.

| Case | Simulation Runs | Ships | Initial Radius (NM) | | Initial Speed (m/s) | | Minimum Initial Distance (NM) |
|---|---|---|---|---|---|---|---|
| | | | Min | Max | Min | Max | |
| 1 | 1000 | 3 | 3.2 | 3.8 | 5 | 10 | 1 |
| 2 | 1000 | 3 | 3.2 | 3.8 | 5 | 10 | 0.8 |
| 3 | 1000 | 3 | 3.2 | 3.8 | 5 | 10 | 0.5 |
| 4 | 1000 | 3 | 3.8 | 3.8 | 5 | 10 | 0.5 |
| 5 | 1000 | 3 | 3.8 | 3.8 | 6 | 6 | 0.5 |
| 6 | 1000 | 4 | 3.8 | 3.8 | 6 | 6 | 0.5 |
| 7 | 1000 | 5 | 3.8 | 3.8 | 6 | 6 | 0.5 |
| 8 | 1000 | 2 | 3.2 | 3.8 | 6 | 6 | 0.5 |

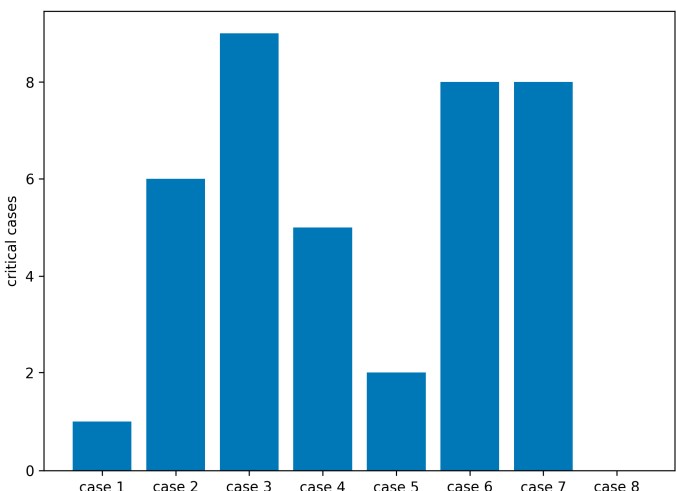

**Figure 14.** Number of definite collisions.

Cases 1, 2, and 3 show the same scenario using different initial distances, where the number of collisions has increased from one to nine by lowering initial distance from one

to 0.5 NM. However, the average minimum distance during the simulation runs does not decrease significantly when lowering to 0.8 NM, but a drop in the average minimum distance is observed at the initial distances of 0.5 NM which is shown in a box blot in Figure 15. Also, the maximum average risk obtained during the simulation runs does not increase significantly, as shown in Figure 16.

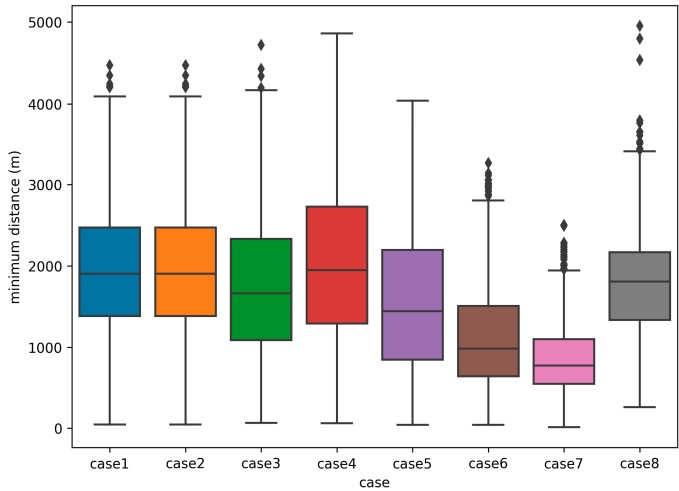

**Figure 15.** Minimum distances between any pair of ships during the simulation runs.

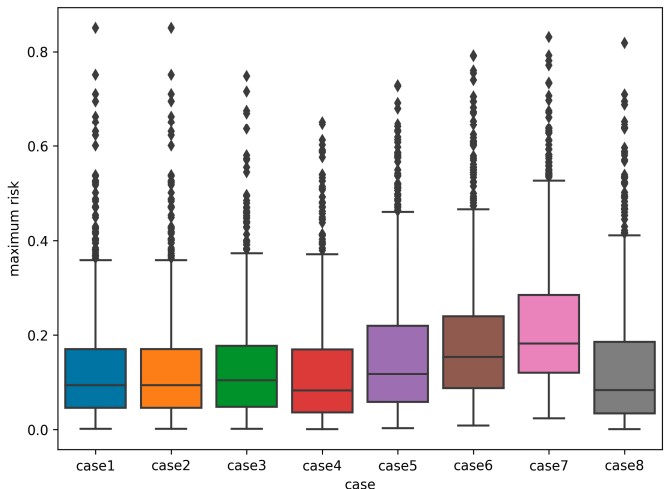

**Figure 16.** Maximum risk during the simulation runs.

When assessing the simulation results, the number of definite collisions was obtained by checking whether the distance between the vessels dropped below 100 m.

## 5. Discussion

The overall results of the simulations indicate that the fuzzy algorithm performs ideally with only two ships. It was important to filter out cases when avoidance would have been impossible due to physical restrictions, so the minimum initial distance was always 0.5 NM or more. As the number of ships increases, an in-depth analysis of critical cases has shown that the avoidance of one target caused a critical condition with another. This suggests that relying solely on COLREGS and avoiding a target with the lowest TCPA or highest risk may result in a crash with a third vessel not identified as dangerous before the maneuver. Therefore, in multi-ship scenarios, there is a need for the implementation of a cooperative collision avoidance algorithm that would prescribe mandatory maneuvers for all involved vessels in areas of dense traffic.

The problem could also be solved by determining a time interval within which the ship could safely execute an avoidance maneuver that would comply with COLREG rules. An example of such a time interval is shown in the encounter with five vessels. In the first part, the model calculates the parameters used to analyze the navigation situation and determine the right of way (Tables 10–12). This is followed by the selection of the most dangerous ship. Its parameters determine the time interval of the relevant decisions.

**Table 10.** Initial parameters.

|  | Own Vessel | Target 1 | Target 2 | Target 3 | Target 4 | Target 5 |
|---|---|---|---|---|---|---|
| C [°] | 82 | 288 | 70 | 262 | 181.5 | 32 |
| V [kn] | 12 | 10 | 27 | 15 | 15 | 23 |
| dt [NM] |  | 7.9 | 3.2 | 9.4 | 8 | 7.8 |
| ωt [°] |  | 85 | 233 | 79 | 32 | 174 |
| Navig. status | Power driven | Power driven | Power driven | Power driven | Power driven | Fishing boat |

**Table 11.** Collision risk assessment with five targets.

|  | Target 1 | Target 2 | Target 3 | Target 4 | Target 5 |
|---|---|---|---|---|---|
| DCPA [NM] | 1.208 | 0.430 | 0.492 | 0.610 | 0.948 |
| TCPA [min] | 21.8 | 12.3 | 20.9 | 23.1 | 26.0 |
| CPA position | negative | Negative | negative | negative | negative |
| RB [°] | 3 | 151 | 357 | 310 | 92 |

**Table 12.** Determination of the right of the way with five targets.

|  | Target 1 | Target 2 | Target 3 | Target 4 | Target 5 |
|---|---|---|---|---|---|
| Sector | IV | II | IV | III | I |
| Vessel's type | Power driven | Power driven | Power driven | Power driven | Fishing boat |
| Nav. Situation | Rule 14 | Rule 13 | Rule 14 | Rule 15 | Rule 15 |
| Vessel with the right of the way | Target ship | Own ship | Target ship | Own ship | Target ship |

According to collision risk assessment (Table 11), there is a danger of collision with ships three and five (DCPA < 1 NM), and at the same time, the ships have the right of way. The model for the most dangerous ship chooses target ship five because it is closer (see Section 3.1—exception 2) and calculates a course change based on the parameters of target ship five, for the time interval of two to 10 min. Table 13 shows the DCPA values for each target in the second and fourth minute of the time delay and the calculated course change.

**Table 13.** DCPA values for each of the target ships in the second and fourth minute of the time delay.

| Time Delay [min] | Course Alteration [°] | New Course [°] | DCPA [NM] | | | | |
|---|---|---|---|---|---|---|---|
|  |  |  | Target 1 | Target 2 | Target 3 | Target 4 | Target 5 |
| 2 | −40.1 | 41.9 | 1.565 | 1.637 | 2.116 | 1.588 | 3.398 |
| 4 | −40.1 | 41.9 | 1.295 | 1.392 | 1.842 | 1.387 | 3.189 |

In the sixth minute of the time delay, the model selects target two as the most dangerous ship based on its distance, since it is 1.68 NM (see Section 3.1—exception 1), and calculates the course change for the sixth, eighth, and 10th minutes (Table 14).

**Table 14.** DCPA values for each of the target ship in the sixth, eighth and 10th minute of the time delay.

| Time Delay [min] | Course Alteration [°] | New Course [°] | DCPA [NM] | | | | |
|---|---|---|---|---|---|---|---|
| | | | Target 1 | Target 2 | Target 3 | Target 4 | Target 5 |
| 6 | −48.6 | 33.4 | 1.500 | 1.219 | 1.985 | 1.562 | 3.759 |
| 8 | −48.6 | 33.4 | 1.175 | 0.948 | 1.656 | 1.319 | 3.466 |
| 10 | −48.6 | 33.4 | 0.850 | 0.677 | 1.327 | 1.076 | 3.173 |

Figure 17 shows the change in the DCPA over time. The appropriate time interval for collision avoidance (considering that target 2 violates COLREG rules) is between the second and the seventh minute, when the safety ship domain is one NM.

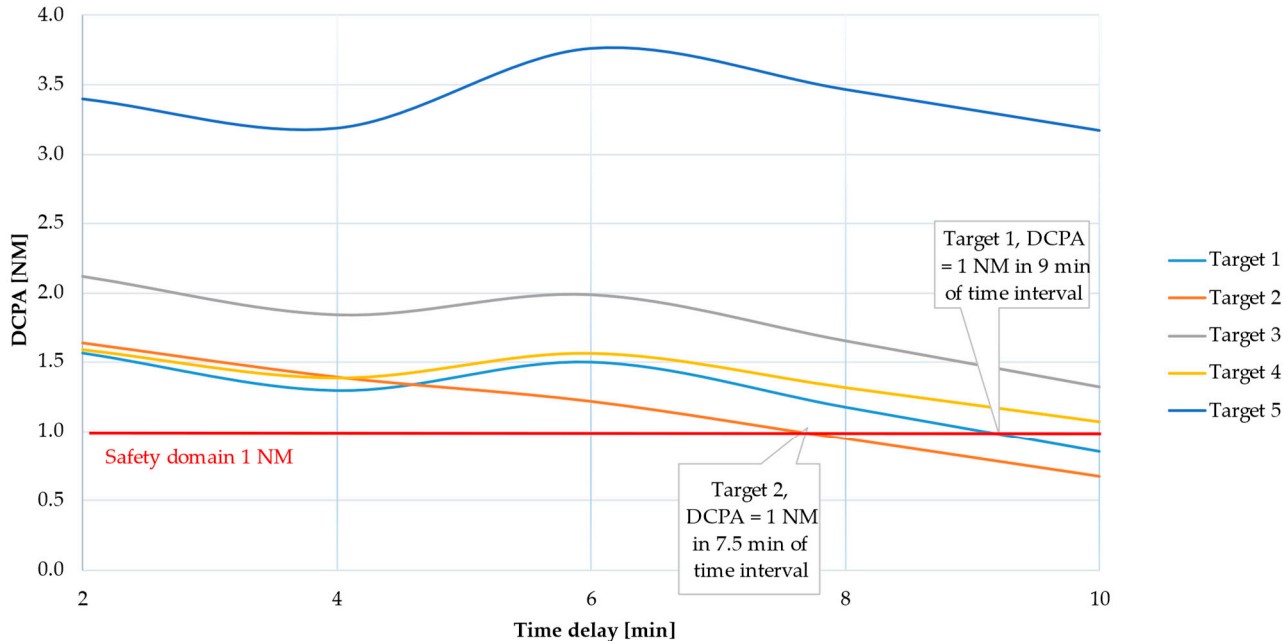

**Figure 17.** Calculated value of DCPA at different time delays.

## 6. Conclusions

According to [45], a majority of collisions on open sea happen at night time, mostly due to poor situational awareness, which is primarily the result of a sole lookout, poorer visibility, lack of communicational skills (misspoken, misread, or misheard information), decisions depending mostly on data obtained from navigation devices, etc. To mitigate predominantly human errors, which account for a substantial 78% of maritime accidents, it is imperative to prioritize research endeavors geared towards the development of sophisticated decision systems. These systems hold the promise of assisting seafarers in making optimal judgments precisely when they are most critical. The veracity and effectiveness of such decisions are intricately linked to the robustness and precision of the data upon which navigation devices operate and are disseminated to end-users. Consequently, decision support systems represent a pertinent steppingstone towards the integration of autonomous vessels, given that their transitional management will predominantly rest upon the expertise and competence of seafarers.

This paper highlights pertinent issues that warrant deeper investigation, notably pertaining to the revision of COLREG (Collision Regulations) rules, which govern the avoidance of collisions at sea, particularly in situations involving multiple vessels. An area of utmost significance in this regard is the examination of the feasibility of determining the right of way from a ship's own perspective. Spatial-Based Trajectory Planning emerges

as a promising avenue for addressing these challenges and should be subject to thorough exploration and research. By delving into this approach, we may uncover valuable insights into enhancing collision avoidance strategies and maritime safety.

**Author Contributions:** Conceptualization, T.B. and B.L.; methodology, T.B. and B.L.; software, T.B and B.L.; validation, T.B.; resources, T.B. and B.L.; writing—original draft preparation, T.B.; writing—review and editing, T.B. and B.L. All authors have read and agreed to the published version of the manuscript.

**Funding:** The authors acknowledge the financial support of the Slovenian Research Agency (research core funding No. P2-0394, Modelling and Simulations in Traffic and Maritime Engineering).

**Institutional Review Board Statement:** Not applicable.

**Informed Consent Statement:** Not applicable.

**Data Availability Statement:** The data presented in this study are available on request from the corresponding author.

**Conflicts of Interest:** The authors declare no conflict of interest.

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
