# Peer review of "A Decision Support System Using Fuzzy Logic for Collision Avoidance in Multi-Vessel Situations at Sea"

_jmse, doi:10.3390/jmse11091819_

Round 1
Reviewer 1 Report
The authors presented the study "A Decision Support System Using Fuzzy Logic for Collision Avoidance in Multi-Vessel Situations at Sea." The introduction has specific problems, which means that it must be corrected as a first phase to give a sense of justification to the study to review the article. Indeed, here are some specific negative points that can be highlighted in the given introduction:
The introduction is written in lengthy and complex sentences, making it difficult for future readers to follow the ideas and concepts presented. This might lead to confusion and hinder comprehension.
The introduction touches on various topics related to collision avoidance at sea, but the structure lacks clear delineation between different concepts and sections. This can make it challenging for future readers to identify the main points being discussed.
The introduction contains many technical terminology and abbreviations related to maritime navigation, rules, and algorithms. This can alienate readers who are not familiar with the field or terminology.
The introduction assumes the reader is already well-versed in maritime regulations, collision avoidance terminology, and navigation scenarios. This can be a barrier for readers who are new to the subject.
While citing relevant literature (overuse of citations:) is essential, excessive citations within the introduction can disrupt the flow of the text and make it feel cluttered.
The introduction touches on the use of advanced technology for decision support in maritime safety; it lacks context about why this topic is important and what challenges it addresses in the real world.
There is inconsistency in the terminology used, such as referring to "collision avoidance" and "maneuvering characteristics" interchangeably without clarifying the relationship between these terms.
The introduction delves into specific details about various authors' work without providing a comprehensive overview of their contributions or significance in the broader context.
The introduction doesn't engage the reader effectively or create curiosity about the research. It lacks a compelling narrative that would motivate future readers to continue reading.
To improve the introduction, consider breaking down the content into smaller, more focused paragraphs, using clear headings, defining technical terms when introduced, providing contextual explanations, and aiming for a more reader-friendly writing style.
In conclusion, It is essential to change the introduction because it gave the sensation that the authors did not follow a correct methodology, and it did not justify this study. For those reasons, I do not recommend the publication of this article.
Smooth transitions between sentences and paragraphs are lacking, making it difficult to follow the logical progression of ideas. The lack of transition phrases can lead to a disjointed reading experience.
Author Response
Dear Reviewer,
Thank you for your careful reading of the manuscript and for your helpful comments and suggestions. We have carefully considered your feedback on the Introduction and agree with your suggested changes. We recognize that the introduction is a critical part of the manuscript because it sets the stage for the rest of the paper and effectively engages the reader. Therefore, we are committed to improving this section to better meet your expectations and improve its overall clarity and impact.
We have divided the introduction into shorter paragraphs, added subsections, and added adequate explanations of certain technical terms that the non-maritime reader can more easily understand and place in the overall context. The writing style has also hopefully improved.
In relation to the English used, we would like to mention that this article was proofread by a professional. Mr Rick Harsh (Master in Fine Arts, in English) is a professional editor, and author and has edited hundreds of papers written in the maritime field. He reviewed this paper, and some changes were made before it was submitted. We cannot, of course, presume that any paper is perfect, but if there are any mistakes in this one or awkward sentences, he would be very happy to revise them if they could be specified.
All changes in the manuscript are marked in red.
Thank you again for your time and expertise in reviewing our manuscript.
Yours sincerely,
The authors
Reviewer 2 Report
The suthors collision aviodance method during multi-vessel encounters based on simulation scenario. This is interesting and useful issue. However, there are still some problems in the manuscript which have to be improved before acceptance.
1. I suggest the unit of "nautical miles" for distance be expressed with full name but not "nm", which is often used for "nano meter" for measurement field.
2. The formula and data used in the manuscript are not eough, which make it difficult to understand the detail of the method.
3.How to calculate the risk of collision? I did not find the information about it in the manuscript.
4. The method and model used in the manuscript were not very clear. There are too much qualitative description,but not mathmatic decsription. The process of modeling was absent.
5. The conditions about the simulation scenarios were not provided clearly.
6. If possible, the experiments should be designed and carried out to verify the method proposed in the research. Just a simple simulation scenario does not prove the feasibility of the method.
Author Response
Dear Reviewer,
We thank you for your careful reading of the manuscript and helpful comments and suggestions. We have made revisions in accordance with your comments and suggestions, as described below.
Comment 1:
It was changed as suggested. We changed to NM, which is an internationally recognized symbol in maritime studies.
Comment 2 & 4:
As was stated in the sections "Introduction" and "Simulations", this article is a continuation of an earlier article in which the decision-making model was explained in detail. In this article, certain parts are summarized to give the reader a general framework of the model, but we have not repeated the formulas that have already been presented. Nonetheless, we have added additional formulas to clarify the nature of the used models.
Comment 3:
The encounter risk was calculated indirectly from DCPA and TCPA. Even though the formulas are generally known and used, we have added them to make the paper more readable. This was added in the equations 5 and 6. The encounter risk used for simulation scenarios is a combination of the two parameters as shown in the equation 11.
Comment 5:
We have provided ship data, however, we haven’t mentioned meteorological conditions that we set as neutral during the simulation in order to exclude this impact which would widen the scope of the paper even further. We, however, plan to analyze that in future work.
Comment 6:
Before actually testing the method, our approach was to simulate thousands of random scenarios to identify potential weak points. When such testing is sufficient, we will proceed to the evaluation of real-world scenarios from AIS trajectories and further to planned experiments.
Thank you again for your time and expertise in reviewing our manuscript.
Yours sincerely,
The authors
Reviewer 3 Report
Topic is original in the field.
Conclusions are consistent.
References are appropriate
Author Response
Dear reviewer,
we thank you for your support and are very grateful to you for accepting our manuscript.
Yours sincerely,
The authors
Round 2
Reviewer 2 Report
I have no extra comment.